# B-Cell Memory Responses to Variant Viral Antigens

**DOI:** 10.3390/v13040565

**Published:** 2021-03-26

**Authors:** Harry N White

**Affiliations:** Department of Biosciences, University of Exeter, Geoffrey Pope Building, Stocker Road, Exeter EX4 4QD, UK; h.n.white@exeter.ac.uk

**Keywords:** virus mutation, antibody, immune memory, vaccine, dengue, influenza, coronavirus

## Abstract

A central feature of vertebrate immune systems is the ability to form antigen-specific immune memory in response to microbial challenge and so provide protection against future infection. In conflict with this process is the ability that many viruses have to mutate their antigens to escape infection- or vaccine-induced antibody memory responses. Mutable viruses such as dengue virus, influenza virus and of course coronavirus have a major global health impact, exacerbated by this ability to evade immune responses through mutation. There have been several outstanding recent studies on B-cell memory that also shed light on the potential and limitations of antibody memory to protect against viral antigen variation, and so promise to inform new strategies for vaccine design. For the purposes of this review, the current understanding of the different memory B-cell (MBC) populations, and their potential to recognize mutant antigens, will be described prior to some examples from antibody responses against the highly mutable RNA based flaviviruses, influenza virus and SARS-CoV-2.

## 1. B-Cell Memory and Its Cross-Reactivity

The development of immune memory is a dynamic process involving the formation over time of memory B- and T-cell populations at different sites in the body. The magnitude and balance of developmental outcomes of these populations, and the strength of memory-based protection afforded, are variable and depend on the pathogen or immunization strategy [1]. The formation of B-cell memory has been covered in depth in excellent recent reviews [1,2,3,4,5].

Prior to antigen exposure the recognition capacity of B-cells for any particular antigen is manifest on a small fraction of the naïve B-cell population [6]. The whole naïve B-cell population, which re-circulates through the blood and secondary lymphoid tissues [7], expresses a vast repertoire of different antibodies formed from the random and imprecise joining of the gene segments that form the antibody gene of any particular B-cell [8].

Antibody responses are initiated principally in the secondary lymphoid tissues, such as the spleen and lymph nodes, when rare antigen-binding B-cells come into contact with antigen. To increase the chance of this happening, after pathogen exposure, these tissues are efficient at trapping and displaying intact antigen either on specialized macrophages in the lymph node sub-capsular region [9] or as opsonized complexes on follicular dendritic cells located in the centre of the lymphoid follicle [10,11]. In concert with this, circulating mature naïve B-cells are continuously recruited from the blood and traffic into the follicles where the antigens are concentrated [12].

After encounter with antigen, B-cell receptor (BCR) signaling leads to activation of the B-cell and the internalization, processing and presentation of antigen-derived peptides on MHC class II molecules [13,14]. Upon activation B-cells then up-regulate chemo-attractant receptors CCR7 and EB12 which stimulate migration of B-cells to the border between B-cell follicles and the T-cell zone [15] allowing them to engage with cognate antigen-primed T-cells and get T-cell help [16] and proliferate.

Around two days after activation a number of B-cell fates become possible. B-cells can develop directly as an extra-follicular (EF) response into principally IgM-expressing memory cells with low levels of antibody somatic hypermutation (SHM), or short-lived antibody-producing plasmablasts of IgM and switched isotypes [17,18,19,20]. At the same time B-cells, with T-cells, also migrate into the B-cell follicles, continue proliferating and differentiate into germinal centre (GC) B-cells, initiating the GC reaction. The exquisite cell biology of the GC reaction has been reviewed excellently elsewhere [21,22]. Briefly, during the GC response, which peaks at around two weeks after antigen exposure, B-cells undergo sequential rounds of proliferation, antibody gene SHM and selection by T-cells which have differentiated into T-follicular-helper cells (TFH). Through this dynamic process B-cells undergo affinity maturation of their BCRs a process that can produce antibodies with sub-nanomolar affinities for antigen [23]. The GC reaction continues for a variable time from a few weeks to several months depending on the nature and complexity of the antigen and any adjuvants [18,24,25].

In considering the potential of B-cell memory to mount cross-reactive responses against variant viral antigens it is important to clarify what is meant by cross-reactive.

Viruses are antigenically complex, being comprised of multiple proteins each having multiple antibody binding sites (epitopes), so an antibody response can consist of many antibodies against many different antigens and epitopes. A definition of antibody memory recognition of mutant antigens needs to separate genuine antibody cross-reactivity (continued recognition of a mutant epitope by promiscuously binding or alternate antibodies), from a shift in focus to more conserved epitopes, and antigens, targeted by distinct sets of antibodies.

Recent studies on memory responses to variant antigens have not made this distinction properly, having used antigens with many epitopes which were also widely different, from different viral strains or escape variants, meaning most epitopes would have had several mutations [25,26,27]. In such situations any ‘cross-reactivity’ will be mostly to alternate more conserved epitopes or antigens, as is typical after heterologous viral infections [28]. In this review we are focusing on genuine cross-reactivity, the continued recognition of mutated epitopes, and how MBCs and their antibodies might enable this.

Throughout the GC response both MBCs and long-lived plasma cells (LLPC) are produced, although it is now apparent there are differences in the rates of production and the selective forces driving these two populations over time that have an impact on the properties and antibody affinities of the resultant memory and plasma cells.

It was initially thought that B-cell memory was largely comprised of B-cells expressing high affinity class-switched/IgG antibodies with high levels of SHM [29,30]. New methods of analysis and the use of transgenic mouse models, however, have subsequently shown that the B-cell memory compartment is very diverse and includes IgM-expressing B-cells with low levels of SHM [24,31,32]. More recent studies have consolidated this view and provide a clearer picture of the development and structure of the B-cell memory compartment in rodents.

Long-lived memory B-cells are made predominantly in the early GC, and many even prior to initiation of the GC reaction [18]. Whilst these early formed cells are predominantly of IgM isotype with low levels of SHM [18,20,26] IgG+ memory B-cells can be formed in a GC independent manner, with low levels of SHM [17]. As the GC response continues, memory B-cells with higher levels of SHM and class switching, are made but in reducing numbers. A key recent study [20] confirms that the great majority of GC-derived memory B-cells express low affinity BCRs, although there is a lesser population of memory B-cells with high affinity for the priming antigen, likely being the population historically associated with the memory response after secondary challenge with an identical antigen.

This study also shows that most MBCs produced later in the GC response have still not increased levels of SHM or antigen affinity, or proliferated significantly compared to plasma cells produced at the same time. Importantly, this shows that the GC reaction is focused on producing larger clones of high affinity antibody-forming cells (AFCs) whilst in parallel auditing the bulk of MBCs to maintain diversity and lower affinity.

Although the diverse structure of the memory B-cell compartment is now better established, the developmental potential of the different cell populations is less understood. While the smaller, high affinity, class-switched memory B-cell population is thought to have a strong bias toward immediate differentiation into plasmablasts and then plasma cells, so supporting memory responses against unaltered antigens, and also replenishing LLPC-based protective immunity [26,32], the function of the ‘less mature’, clonally diverse, low affinity IgM-expressing, memory population is less clear.

The initial discovery of this population led to speculation that this diversity would allow recognition of antigen variants, and so enhance memory responses to mutated pathogens, either by rapid differentiation of memory cells into antibody-forming cells (AFCs) or by the re-starting of the GC reaction, so providing an advantage over naïve B-cell-derived responses [18,24,32,33].

There is good evidence showing the less mature memory B-cell population readily re-instigates GC responses after secondary antigen exposure [24,32,33] and indeed that this occurs against variant antigens [25].

More recently, however, it has been shown in rodents that GC re-entry by memory B-cells may not be a strong effect, that GC responses after secondary exposure are dominated by newly recruited naïve B-cells [20,26] and that this balance is not altered when the secondary challenge is with a variant antigen [26,27].

There is an important caveat to this latter observation, however, that the level of variation in the second antigen was ‘high’, being from a different influenza strain, H1 California/07/09 or H5 Indonesia/05/05 after primary infection with H1 Puerto Rico/08/34, [26], or from epitopes containing key known escape mutations in the West Nile virus envelope protein, K307E and T330I, that abolish neutralizing antibody binding [27], and so may be beyond the high-affinity binding capacity of many of the pre-existing memory cells recruited by the priming antigen. These issues are discussed further later.

Even single amino acid substitutions in epitopes can reduce or abolish binding of high-affinity antibodies [34,35] including by several at once [36,37]. As this is the level at which viral mutation most often manifests [38], likely resulting in a range of disruptions to epitope structure, it will be important in the future to clarify the developmental outcomes of MBC cross-reactivity to these different levels of variation.

MBC diversity has long been proposed to support cross-reactivity with the presence of B-cells with low SHM with ‘promiscuous’ antigen-binding sites, allowing binding of a wider range of related epitopes [18,24,39,40], although until recently this issue was poorly understood in terms of MBC antibody biophysics.

A recent study [41] demonstrates one mechanism through which MBC antibodies may be more adaptable to the levels of variation seen with viral point mutations or seen between ‘nearly conserved’ epitopes present on more distantly related viral strains. After analyzing several hundred recombinant antibodies cloned from plasmablasts and MBCs induced by influenza infection or vaccination, a minor subset of haemagglutinin (HA)-specific antibodies was shown to be polyreactive as defined by reactivity with a panel of diverse antigens. The majority of these HA-specific polyreactive antibodies bound the most conserved—but still variant—regions of HA, principally to the HA stalk region but also to the more conserved RBS region of the usually variable HA head. This suggested that polyreactivity is a common feature of antibodies that can bind related epitopes. In addition to this, it was shown these antibodies are induced preferentially after secondary exposure to novel viral strains. The strength of polyreactivity of particular antibodies correlated with the strength of specific antigen binding and molecular dynamics simulations showed the antigen-binding sites were more flexible suggesting how the antigen-binding region could adapt and stabilize binding to altered epitopes. The polyreactive antibodies tended to have lower levels of SHM [41] and have previously been shown to have longer more hydrophobic or positively charged antigen-binding regions [42,43,44] altogether providing support for the proposal that a diverse memory antibody pool can provide antibodies with properties conducive to cross-reactivity.

In addressing the question of whether binding by less mutated antibodies might be more promiscuous it has previously been shown that both germline and mutated antibodies can bind multiple ligands, often through conformational isomerisation [45,46]. One of these studies [46] also showed promiscuity was mediated through specific, e.g., hydrogen bonded, interactions rather than a more general hydrophobic stickiness particular to less mutated antibodies. The specificities analyzed, however, were to widely different antigens, suggesting this type of interaction might occur rarely between more related epitope variants. It is perhaps the case, therefore, that antibodies can bind multiple ligands in different ways, by being flexible and perhaps polyreactive, by having hydrophobic binding sites or by fortuitously having particular high strength interactions available; the chance of all such properties being increased in a more diverse memory pool that has not been selected on the basis of high affinity to the priming antigen.

The large clonal diversity of memory should also support cross-recognition by chance, on top of that potentially afforded by promiscuous antibodies. The antibody region that forms the contact with the epitope is called the paratope. Antibodies with different paratopes can bind the same epitope but with different reactivity to epitope variants [47]. It seems plausible, therefore, that from within the greater diversity of antibody memory there could be many alternate specificities, not manifest in the more restricted primary or secondary AFC responses to a particular antigen [26,48], that can mount memory responses to variant antigens.

Whilst as previously described, antigens often have many epitopes and antibody responses produce many antibodies against these, the distribution of the response against different epitopes is often not uniform. Antibody responses can become focused on particular epitopes in the complex process of immunodominance [49,50,51]. Further, it has been recently demonstrated using influenza heamagglutinin (HA), that has five well-defined antigenic regions, that the immunodominance hierarchy of responses against these regions shifts as the response matures, varies depending on whether it was induced by infection or different routes of vaccination, and varies between strains of mice—suggesting similar would occur within different humans [50].

An important effect of epitope immunodominance is that, at least transiently, selection for viral escape mutations becomes focused on a single or fewer epitopes, increasing the chance of mutational escape. Whilst the presence of high-affinity antibody against the immunodominant epitope can suppress the secondary response to that epitope [50], suggesting a form of feedback, the overall regulation of immunodominance is incompletely understood, although it can be manipulated [51]. Thought to be a least in part a viral immune evasion strategy [51,52], it is also the case that the AFC population is far less clonally diverse than the MBC population, perhaps making the AFC response more vulnerable to flipping into immunodominance during the complex dynamics and inter-clonal competition of a developing antibody response.

## 2. Viral Point Mutations and B-Cell Memory Responses

RNA viruses such as influenza and dengue have high mutation rates during genome replication for a number of reasons including through the use of RNA polymerases that lack proof-reading ability and have low fidelity [38,53,54]. Considering the rate of substitutions per nucleotide per cell infection as the mutation rate for such viruses, shows rates of approximately 10^−4^ for dengue and 2 × 10^−5^ for influenza, 80% of which will be substitutions with the remainder being insertions and deletions [55]. For dengue and other flaviviruses, with genomes of approximately 10 kb, this equates to about one DNA substitution, most of which will result in amino acid replacements, per genome per cell infection. As the rate of virus production in dengue and influenza can reach several thousand per infected cell [56,57], an infected cell could, theoretically produce thousands of virus variants, with amino acid replacement at many positions in viral proteins. There are obvious constraints on this process, not least that many mutations are deleterious and many others will introduce stop codons. This mutability, however, suggests that during active infections replicating populations of such viruses are extremely efficient at exploring protein sequence space, as a swarm of virus variants, in a process that searches for increases in viral fitness whether it be through improved transmissibility or immune escape. As such these viruses present a formidable challenge to the immune system.

As mutations are often deleterious, it follows that there is an optimum mutation rate, beyond which genomes become too often dysfunctional through premature stop-codons or loss of function in encoded proteins. This would suggest that viruses with smaller genomes might be able to sustain higher mutation rates, due to the lesser chance of a strongly deleterious mutation per genome, and there is indeed a negative correlation between viral genome size and mutation rate [55]. As an example from this report, mouse hepatitis virus, a coronavirus, has a genome around three times larger than dengue virus and a mutation rate 30-fold less.

Studies on the sequence variation of dengue and influenza viruses reveal that negative selection occurs, reducing the levels of many variants both within hosts and after transmission into populations [58,59], a reflection of the selective forces that both act on the many deleterious viral mutations, and that drive the evolution of perhaps rarer immune escape or enhanced transmissibility variants.

Of the mutations that alter viral antigenicity, without otherwise impacting viral fitness, what is poorly understood is the proportion and characteristics of those that are successfully dealt with quickly by immune responses. Whilst it is reasonable to assume that many mutations in viruses, especially those with modest effects on antigenicity, will still be recognized by MBC antibodies, whether they appear during an ongoing response, or as part of a new epidemic, the scale or breadth of such mutation and then recognition is by definition hard to detect. It is only the successful, likely ‘stronger’, mutations that manifest as escape variants, likely also in the context of an immunodominant antibody response.

A seminal study on a viral envelope protein mutant showed both that the MBC compartment had specificity for the mutant when the LLPC population had lost it, and demonstrated the enabling effect of an immunodominant neutralizing antibody response on viral escape [60]. Strongly neutralizing antibodies against West Nile virus, a flavivirus related to dengue virus, bind the lateral ridge epitope of the viral envelope protein [61] and after viral infection most LLPC antibodies are focused on this epitope [60]. Serum from wild type virus-infected mice was much reduced in antigen binding, virus neutralization and inhibition of viremia when a particular mutation was present on the epitope. In vitro-stimulated MBC, however, produced antibodies that bound the mutant protein, including particular antibodies that bound at higher affinity than to the wild type protein, and polyclonally induced MBC supernatants neutralized the wild type and mutant virus equivalently. This clearly demonstrated that the MBC compartment, known to be diverse, contains specificities for epitope variants that the primary response by AFC does not have.

This study also again highlighted the effect of antibody immunodominance on viral escape, something that has also been observed in humans infected by influenza virus point mutants or drift variants [36,52]. Completely epitope-focused antibody responses, however, may not be a pre-requisite for viral immune escape. As virus neutralization depends on exceeding a certain threshold proportion of total epitope sites occupied by neutralizing antibodies [62,63], mutation at one epitope could just tilt the net balance away from neutralization.

A recent report extended observations on cross-reactivity of MBC, demonstrating that after secondary exposure to an escape variant, or a related viral strain, the cross-reactive MBC response did not undergo further affinity maturation in GCs [27]. Further it was again shown that the MBC population largely had low affinity for the priming antigen, consolidating other recent studies [20,64] indicating that selection of high affinity cells either before or during the GC reaction drives B-cells to the plasma cell fate and survival permission of lower affinity cells is the mechanism that maintains diversity in the MBC compartment.

## 3. SARS-CoV-2 Variants and Antibody Immunity

This review has been produced during the ongoing pandemic of SARS-CoV-2. After just over a year of the pandemic it has become obvious there is a serious problem with the emergence of viral envelope spike protein variants that are impacting virus transmissibility and vaccine- and infection-induced antibody based immunity. Current SARS-CoV-2 mutations are collected in two areas, the N-terminal domain, where they reduce neutralizing antibody binding, and in the RBD where they have a complex effect on transmissibility and immune escape by increasing RBD affinity for its ACE2 receptor and reducing binding of neutralizing antibodies [65].

Of particular current concern are the B.1.351 ‘South African’ variant and the P.1 ‘Brazil’ variant, as there is now strong evidence their mutations enable escape from neutralization, as measured in vitro, by convalescent plasma and monoclonal antibodies [66], and serum from current vaccinees [65]. The B.1.351 variant shows a reduction in neutralization by current vaccine sera of around 7- to 9-fold.

Due to the globally high rates of infection and because the virus has a zoonotic origin and so will be adapting to human hosts, evolution may have been expected to occur, perhaps initially rather rapidly. The speed with which this can occur is dramatically illustrated by the dominant appearance of an immune escape variant in an immune-compromised patient treated with convalescent sera [67].

The reduction of in vitro neutralization by serum or antibodies derived from ancestral/Wuhan-type virus infections is a reflection of immune escape from AFC derived antibodies present after a primary infection as has been observed with flavivirus variants [60]. What is currently less scientifically understood is the in vivo course of secondary SARS-CoV-2 variant infection and whether cross-reactive memory responses may offer enhanced protection compared to non-immune individuals. Unfortunately, some epidemiological evidence points the other way suggesting that equivalently severe secondary infections by variants can occur with marked re-infection of majority seropositive populations by the P.1 variant in Brazil [68] and the observation that prior seropositivity had no impact on infection rates by the B.1.351 variant [65].

A recent study on anti-SARS-CoV-2 B-cell memory responses reports that while serum antibody titres wane, memory responses remain strong up to six months after infection and, further, continue to improve neutralization potency and breadth [69]. Such a process is consistent with continued GC activity perhaps enhanced by the observed continued presence of virus in the small intestine. From these patients a panel of temporally and clonally related antibodies were cloned from memory B-cells, and analysed for binding to RBD variants. It was shown for many of the antibodies that those derived from later in the infection showed improved binding to variants, likely from the ongoing GC activity. It is notable that reactivity to the E484K variant, present in pandemic variants associated with immune escape, that this effect was not strong. These highly novel findings suggest that mutations such as E484K enable at least partial escape from memory antibody derived immunity, consistent with the observations from Brazil [68], and, further show that the virus has been very efficient at evolution.

Interestingly, studies evolving SARS-CoV-2, or its spike protein, in vitro with selection by mAbs or serum against earlier strains, has produced a similar set of mutations to those circulating now [70,71,72]. This suggests we may have already arrived at the mutations that are most efficient at evading immunity and memory induced by earlier strains, although this situation will likely change when the population is exposed through vaccination or infection to the variants themselves.

## 4. Concluding Remarks

The MBC compartment is known to be diverse, contain largely low affinity B-cells and contain cross-reactive specificities for antigen variants that are not present in the AFC compartment which is less clonally diverse. Whilst some MBC re-enter the GC in response to a variant antigen the bulk of the MBC response is currently thought to involve immediate differentiation into AFCs, the important advantage being a more rapid humoral antibody response to viral variants compared to a naïve response. The variant antigens analyzed to date, however, have either had ‘strong’ mutations or have come from different strains so have mutations in the epitopes that may be too different to efficiently, cross-reactively, stimulate MBC into the GC pathway. This proposal is further supported by the observation that productive engagement in the GC reaction is dependent on antibody affinity [20], and so less extreme variant epitopes may be more likely to engage MBC antibodies at sufficient affinity to drive MBC into the GC reaction.

In further support of this, an important recent study in humans after influenza vaccination showed significant contribution of MBCs to GC responses [73]. This study showed that a large proportion of GC B-cells recruited to GCs after immunization had high levels of SHM and extensive clonal overlap with rapidly induced plasmablasts, indicating a memory origin. These B-cells had antibody specificity for the more conserved epitopes of HA supporting the proposal that more closely related antigen variants efficiently stimulate GC re-entry by MBC. In addition to this, the study found there were significant numbers of GC B-cells with low levels of SHM after vaccination, clonally unrelated to contemporary plasmablasts, and with specificities for the more variable, strain specific, epitopes on the immunogen. These cells likely represented a de novo primary response to epitopes too variant to stimulate MBC. Whilst it is possible the contribution of MBC to the GC response in this study may have been boosted by residual GC activity in the draining lymph nodes [26], the study clearly shows that naïve- and MBC-derived GC responses can occur efficiently in parallel, and perhaps with a balance of contributions based on which population has higher affinity for the particular epitopes available.

Considering these studies, there is the interesting question of why MBC participation is not greater after re-challenge with an identical antigen, which should bind MBCs most efficiently. This is likely at least in part due to the presence of high affinity antibodies against the priming antigen that inhibit MBC GC participation and this has been demonstrated [32]. Further, as most MBC have not been selected for high affinity to the priming antigen, the affinity of serum antibodies may not need to be particularly high to have this effect. It seems plausible, therefore, that MBC participation in GCs may occur most efficiently when antigens have mutated sufficiently to escape binding by AFC-derived antibodies, likely aided by AFC immunodominance effects, but are still recognized by MBC due to their antibody promiscuity or paratope diversity as discussed above.

At the limits of recognition by B-cell memory of infecting viruses, certain phenomena occur that can have a negative impact on immunity. Secondary dengue virus infection often occurs during an epidemic wave of one of four strains of virus with envelope proteins that can have as little as 65% amino acid identity with a previously infecting strain [74], and similarly dramatic changes can occur in the influenza virus surface proteins after antigenic shift caused by genome re-assortment prior to infection of humans [75].

There is likely to be a limit to alteration in the structure of an epitope beyond which it will not be recognized effectively by any MBC. Such altered epitopes, however, are often expressed on newly infecting virus particles with other epitopes and antigens that may be more conserved and so are still recognized by other MBC, stimulating a shift in any secondary response away from the antigen focus of the primary response. Many such epitopes or antigens are less neutralizing and in the case of dengue, may seriously exacerbate infection through the promotion of antibody-dependent enhancement (ADE) of virus infection by targeting virus to Fc receptor-bearing cells [28,63].

A related issue is that of original antigenic sin (OAS). Secondary infections by a variant virus can induce MBC-derived antibodies with much higher avidity for the original infecting virus. MBCs have a lower activation threshold than naïve B-cells [76,77], and so even weakly binding antigens can trigger secondary AFC responses against the variant epitope. OAS is a complex phenomenon that can occur after both influenza and dengue infections [78,79]. There may be benefits in the rapidity of the OAS secondary response, and it has been excellently reviewed elsewhere [79]. In dengue, however, OAS has the potential to cause major problems as the weakly cross-reactive, low avidity OAS antibodies may fail to neutralize but can still cause ADE [80]. These two complex phenomena demonstrate limitations of B-cell memory responses, showing what processes remain after the effectiveness of genuine cross-reactivity has diminished.

In seeking to understand how to improve vaccines to induce antibody-based immunity more protective against mutable viruses, two broad approaches are being followed. There is intensive research on inducing memory against more conserved, harder to target, neutralizing epitopes [81,82]. In concert with this, advances are being made in the basic biology of MBC that should allow vaccine induction of more genuinely cross-reactive MBC recognition capacity as discussed in this review. Bringing these two areas together, there has been promising recent work with the use of mosaic antigens that induce more broadly cross-neutralising responses. Protective immunity is often induced by viral infection but such antibody responses are usually focused on immunodominant, strain-specific epitopes [63,83]. This increases the challenge of inducing more cross-protective responses against conserved but less dominant epitopes. By constructing a nanoparticle antigen with mosaic display of influenza HA RBS domains from two divergent H1N1 strains, Kanekiyo et al. [83] were able to induce broadly neutralizing antibody responses against multiple H1N1 strains. Clever spacing of the heterotypic RBS domains on the nanoparticle meant antigen binding was strongly biased toward antibodies that could bivalently bind both types of RBS. The authors also showed that the response was not focused on the usually immunodominant and strain specific epitope sequences around the RBS, indicating how undesirable immunodominance can be overcome. Such an approach offers great promise, particularly when the properties of memory induced by mosaic antigens is better understood, and combined with likely progress in understanding what factors increase re-engagement of memory cells with the GC reaction, and how antibodies can be promiscuous, there is good reason to be optimistic.

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
