# Peer review of "B-Cell Memory Responses to Variant Viral Antigens"

_viruses, 2021, doi:10.3390/v13040565_

Round 1

Reviewer 1 Report

This is an excellent and well written review summarizing the latest concepts in B cell memory and antibody responses.  The article is timely, curent, and relevant to viral epidemics and vaccines. 

Author Response

Thank you for your positive and encouraging comments. You did not request amendments so there are none for you.

Reviewer 2 Report

This is a well-written review to introduce B cell memory and related response to viral mutation. It is well organized, informative and will benefit the research community of anti-viral immunology. 

Author Response

(The authors gave the same response as above.)

Reviewer 3 Report

The review ‘B-cell memory responses to variant viral antigens’ submitted to Viruses aims to describe the conflict between the mounting of a highly specific and effective immune response including an antigen-specific immune memory upon virus infection and the pathogen’s ability to escape this memory by mutation of their antigens. The author highlights the potential and limitations of antibody memory to protect against virus variants, and the potential of different memory B-cells populations, a topic particularly of value in the middle of an ongoing SARS-CoV2 pandemic and the repeated rise of new strain variants.

Major comments:

Overall, the review is concise, balanced and very well written. However, being not a full-fledged immunologist (and the manuscript being a review article in Viruses) the reviewer would welcome a few figures or flow charts depicting for example possible fates of B cells or the varying populations of B cell memory cells. The author also included ‘coronavirus’ as one of his key words, however there is no strong mention of coronaviruses. The reviewer strongly recommends to include a section about coronaviruses, due to obvious current interest in the virology field.

Minor comments:

Page 3, line 122-125 it would be helpful to be more specific -  what are the different influenza strains or key known escape mutations?

Page 7, line 329 elaborate on mosaic antigens in vaccine development as this is an approach trying to alleviate several of the issues that were raised in the review
